# Synthesis of inter-[60]fullerene conjugates with inherent chirality

Yoshifumi Hashikawa [1] ✉, Shu Okamoto[1] & Yasujiro Murata [1] ✉

Coalescence of [60]fullerenes potentially produces hypothetical nanocarbon assemblies with non-naturally occurring topologies. Since the discovery of [60]fullerene in 1985, coalesced [60]fullerene oligomers have only been observed as transient species by transmission electron microscopy during an oligomerization process under a high electron acceleration voltage. Herein, we showcase the rational synthesis of covalent assemblies consisting of inherently chiral open-[60]fullerenes. The crystallographic analyses unveiled double-caged structures of non-conjugated and conjugated inter-[60]fullerene hybrids, in which the two [60]fullerene cages are bounds to each other through a covalent linkage. The former one further assembles via a hetero-chiral recognition so that four carbon cages are arranged in a tetrahedral manner both in solution and solid state. Reflecting radially-conjugated double $\pi$-surface nature, the inter-[60]fullerene conjugate exhibits strong electronic communication in its reduced states, intense absorption behavior, and chir-optical activity with a dissymmetry factor of 0.21 (at 674 nm) which breaks the record for known chiral organic molecules.

[60]Fullerene ($C_{60}$) is the most abundant molecular carbon cluster which possesses twelve pentagonal rings surrounded by hexagons, thus causing a topological ring closure into an icosahedron[1]. Inspired by its characteristic physical nature such as electron-accepting[2], superconducting[3,4], and ferromagnetic behavior[5], a variety of hypothetical carbon-based polyhedrons have been proposed, since the early 1990s, by rearranging atomic coordinates of $C_{60}$ and/ or fusing multiple molecules of $C_{60}$[6]. In the latter case, cross-linkages potentially produce covalent assemblies of carbon clusters with non-naturally occurring topologies as found in fancy mesoporous fullerene sponges[6,7] and multidimensional polycrystalline fullerites[8,9]. Hence, the coalescence of two or more fullerene cages has been a topic of great interest (Fig. 1). As structurally well-defined covalent assemblies, bis(aza[60]fullerenyl) $(C_{59}N)_2$[10,11], [2 + 2]-cycloadduct $(C_{60})_2$[12], and their endohedral congeners[13,14] have been found in succession (Fig. 1a). The cross-linkage is usually constituted by an $sp^3$-hybridized spacer which, however, disconnects the $\pi$-conjugation between the two cages, thereby observing negligible or faint intercage interaction at the very best. Fully conjugated inter-[60]fullerene allotropes are, in contrast,

observed only as transient species in an oligomerization process of $C_{60}$ inside carbon nanotubes[15–17]. At the present time, a lack of methodologies has severely hampers the synthesis of long-envisioned, inter-[60] fullerene conjugates in an isolable form, yet leaving its understanding elusive over few decades.

Herein, we showcase the synthesis and solid-state structure of an inter-[60]fullerene conjugate. By the use of an open-[60]fullerene[18,19] as a pairing molecule (Fig. 1b), the full $\pi$-conjugation would be realized along with the two cages in a radial manner which enables an effective conjugation of the inner and outer $p_z$ lobes (Fig. 1c). This is reminiscent of a twist in-plane conjugation in lemniscular aromatics[20]. Different from known [60]fullerene dimers comprised of closed cages[15–17], the conjugation of the two open-[60]fullerenes endows functions such as host–guest complexation as well as intercage electronic communication and chiroptical properties. Our synthetic trials are commenced under three strategies using open-[60]fullerenes **1**–**3**[21] as potential precursors, i.e., Wittig reaction, aldol condensation, and consecutive deoxygenation, the last of which works well to generate covalent assemblies of open-[60]fullerenes (Fig. 1c).

[1]Institute for Chemical Research, Kyoto University, Uji, Kyoto 611-0011, Japan. ✉e-mail: hashi@scl.kyoto-u.ac.jp; yasujiro@scl.kyoto-u.ac.jp

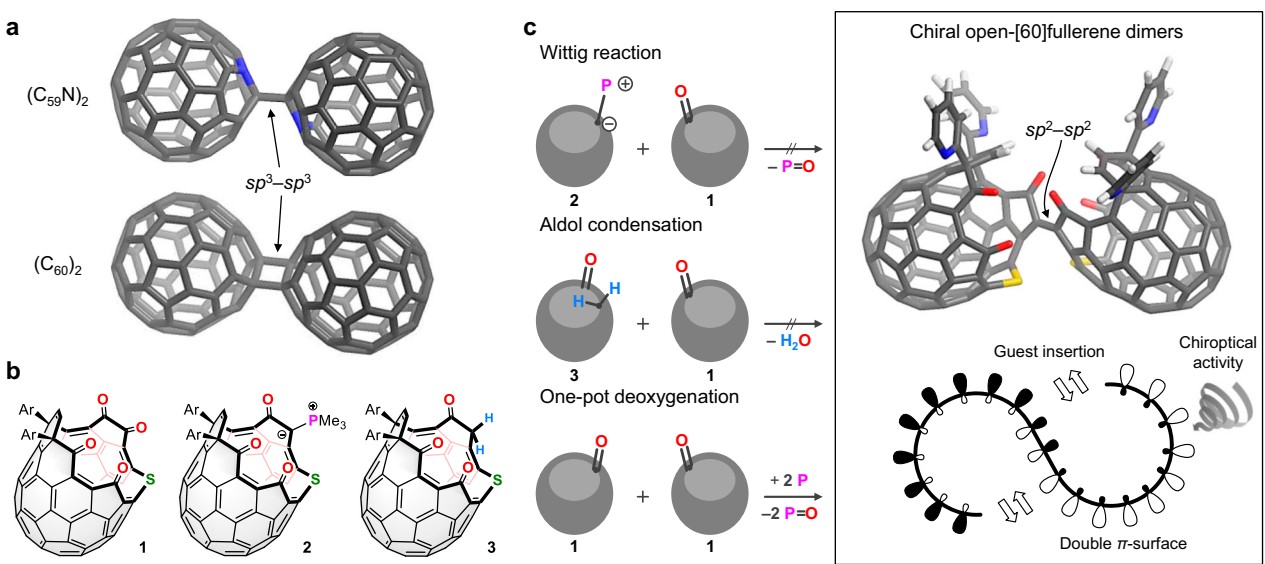

**Fig. 1 | [60]Fullerene assemblies. a** Classical [60]fullerene dimers (gray for C and blue for N). **b** Potential precursor molecules (Ar = 6-*t*-butylpyridin-2-yl). **c** Strategies for the synthesis of chiral open-[60]fullerene dimers with a radial π-conjugation (gray for C, blue for N, red for O, yellow for S, and white for H).

## Results and discussion

### Synthesis and structure

With the former two strategies, the reactions of **1** with **2** or **3** were initially examined and the desired open-[60]fullerene dimer was generated only in a trace amount (Supplementary Fig. 1). Once the third strategy, that is a phosphine-mediated sequential deoxygenation, was adopted, conjugated dimer **4** was obtained in 20% isolated yield (Fig. 2a, b), in which 1-phosphonium-3-oxobetaine[21] might be generated as a key intermediate by the reaction of **1** with the phosphine (Fig. 1c). The molecular ion peak at *m/z* 2236.3428, which is assignable to [(1)$_2$−2 O]$^{\cdot-}$, corroborated the formation of the dimer via deoxygenation. In the same reaction, hydrogenated dimers, **5** and **6**, were also formed in 4 and 5% yields, respectively, in which the hydrogenation is assisted by trimethylphosphine[22,23] so that the conversion of **4** into **5** took place in 49% yield. Considering the inherent chirality of **1**, the two compounds are regarded as homochiral (*rac*-**5**) and heterochiral (*meso*-**6**) dimers which are also describable as ($^{f,s}C$,$^{f,s}C$)-**5**/($^{f,s}A$,$^{f,s}A$)-**5** and ($^{f,s}C$,$^{f,s}A$)-**6**/($^{f,s}A$,$^{f,s}C$)-**6** upon adopting chiral descriptors of $^{f,s}C$ (clockwise) and $^{f,s}A$ (anticlockwise)[24]. Notably, this deoxygenative dimerization is highly homochiral selective. The use of enantiomerically-pure **1** with a $^{f,s}C$-configuration, therefore, improved the yields of ($^{f,s}A$,$^{f,s}A$)-**4** and ($^{f,s}A$,$^{f,s}A$)-**5** in 32 and 13% yields, respectively, while **6** was not formed as a matter of course. Note that the change in stereodescriptor does not indicate a chiral inversion but is because of the reversed priority order of C1 and C2 in both **4** and **5** by losing oxygen atoms. According to theoretical calculations (Supplementary Fig. 29), **6** is more thermodynamically stable than **5**. The crystallographic analysis undoubtedly confirmed a double-caged structure of **4** (Fig. 2c). The two carbon cages are connected by a C = C bond (1.349(7) Å) at a torsion angle of 18.8°, thus covering the orifice by the paring molecule. The chemical species found inside the cages were refined as a disorder of N$_2$ and Ar. In ODCB-$d_4$ (*o*-dichlorobenzene-$d_4$), a water molecule gains entry through the orifice and reaches to an occupation level of 24% at room temperature, being indicative of the flexibility of the double-caged structure despite the seemingly crowded entrance. Within an asymmetric unit, two crystallographically independent molecules were found for non-conjugated dimer **5**, in which the two molecules, i.e., ($^{f,s}C$,$^{f,s}C$)-**5** and ($^{f,s}A$,$^{f,s}A$)-**5**, are assembled as a dimer so that four carbon cages are arranged in a tetrahedral manner (Fig. 2d). This heterochiral recognition is enabled by multiple

hydrogen-bondings complementarily formed between pyridyl protons and carbonyl oxygens on the four cages whose atomic arrangement is a stereogenic element. The ¹H NMR (nuclear magnetic resonance) spectrum of *rac*-**5** in benzene-$d_6$ showed signals of both **5** and (**5**)$_2$ owing to the slow association whereas ($^{f,s}A$,$^{f,s}A$)-**5** showed only single component under the same conditions. This is supportive of a self-sorting via chiral discrimination between ($^{f,s}C$,$^{f,s}C$)-**5** and ($^{f,s}A$,$^{f,s}A$)-**5**, giving a heterochiral assembly both in crystals and solution. The association constant was determined to be $1.11 \times 10^4\,M^{-1}$ in benzene-$d_6$ at 300 K, which corresponds to $\Delta G = -1.43\,kcal\,mol^{-1}$ (Supplementary Figs. 20 and 21).

### Electronic properties

To unveil electronic properties of open-[60]fullerene dimers, we recorded absorption spectra in toluene (Fig. 3a). Different from non-conjugated dimer **5**, the spectrum of **4** could not be described as a simple two-fold absorption of monomer **1**, being suggestive of the effective π-conjugation along with the two cages. The electrochemical analysis of **4** showed clear separation of six one-electron reduction waves, indicating that one-electron injection into one side in **4** electrochemically perturbs the electron uptake behavior of the counterpart (Fig. 3b). This stands in sharp contrast to non-conjugated dimers **5** and **6** which showed three-step two-electron reductions with negligible peak separation[25]. The localized orbital locator (LOL) isosurface (Fig. 3c)[26,27] confirms the π-orbitals delocalized over the entire carbon skeleton through the olefin linkage, where a double π-surface is arranged in a radial manner. Accordingly, the highest-occupied and lowest-unoccupied molecular orbitals (HOMO and LUMO) of **4'** are delocalized within the dimeric structure (Fig. 3d). The longest wavelength absorption band at λ = 685 nm (computed transition energies were scaled[28] by 72%[21].) was assignable to π−π* transition with a large oscillator strength of *f* = 0.2358. Since anionic charge and spin density of (**4'**)$^{\cdot-}$ are delocalized over the entire π-skeleton (Fig. 3e), the well-resolved reduction waves are ascribed to the strong intercage electronic communication through the olefin linkage. This concave−convex conjugation is characteristic to the open-[60]fullerene conjugate whereas it could not be discriminated from concave−concave/convex−convex conjugations for analogous Buckybowl dimers consisting of corannulene[29] or sumanene[30] due to possible bowl inversion as well as rotation even along the olefin linkage.

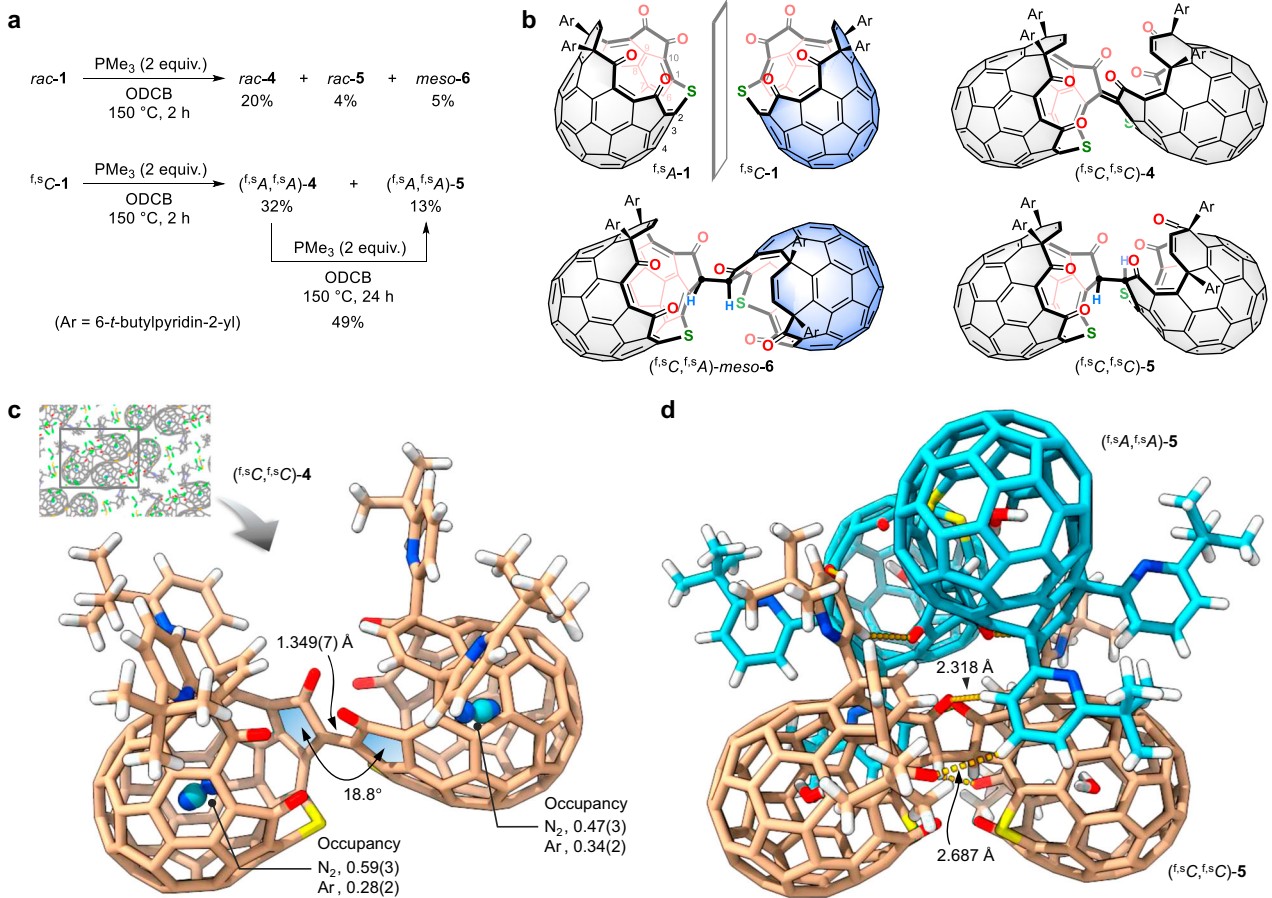

**Fig. 2 | Synthesis and structures of open-[60]fullerene dimers. a** Reaction conditions and structures. **b** Chemical structures of **1**, **4**, **5**, and **6**. **c** Crystal structure of **4** (beige for C, blue for N, red for O, yellow for S, and white for H). The inset represents a packing structure. **d** Crystal structure of **5** (The two independent molecules with different chiral configuration are shown with beige and sky blue for C, blue for N, red for O, yellow for S, and white for H). The solvent molecules are omitted for clarity.

## Chiroptical properties

Open-[60]fullerenes are less-explored chiral chromophores in which their inherent chirality originates from the orifice structures[31–33]. Accordingly, **4** and **5** are regarded as chiral nanocarbon assemblies with a $C_2$ symmetry (Fig. 4a) while **6** is achiral due to the presence of a mirror plane ($C_S$ symmetry). Optical resolution of **4** was achieved by chiral high-performance liquid chromatography, affording two enantiomerically-pure fractions which are identifiable by different retention times of 11.9 and 23.4 min (Fig. 4b). In circular dichroism (CD) spectra, the intense cotton effect with $\Delta\varepsilon$ up to ca. ±300 M⁻¹ cm⁻¹ was observed over a wide range reaching far-red region (Fig. 4c). Judging from the simulated CD spectra (Supplementary Fig. 32), the first- and second-eluted samples are assignable to (f,sC,f,sC)-**4** and (f,sA,f,sA)-**4**, respectively. As a consequence of the reaction using an enantiomerically-pure starting material (Fig. 2a), the origin of the stereogenic element of (f,sA,f,sA)-**4** (fraction 2) was chiroptically traceable to f,sC-**1** (fraction 1) and (f,sA,f,sA)-**5** (fraction 2) (Supplementary Figs. 22–26). The dissymmetry factor, a measure of chiroptical activity, of inter-[60]fullerene conjugate **4** was recorded to be $g_{abs}$ = ±0.210 with an order of 10⁻¹ at 674 nm in toluene (Fig. 4d). This is among the largest ever reported for conventional chiral organic molecules as their $g_{abs}$ values are in general found at an order of 10⁻⁴–10⁻³ [34]. The similarly large values have been reported for hexahydropentalenone (ca. 0.2 supposedly at ultraviolet region)[35] and cycloarylene (0.167 at 443 nm)[36]. Monomer **1** showed $g_{abs}$ = ±0.063 at 648 nm, which is rather smaller than another open-[60]fullerene reported previously (±0.20 at 710 nm)[32]. This might arise from the large l7-atom-ring orifice in **1**

causing a flexibility of the caged structure, which partly breaks radial π-conjugation in solution while the latter is structurally rigid owing to the small 12-atom ring. As a result of an enhanced rigidity by dimerization, **5** attains a twice larger $g_{abs}$ value (±0.063 at 648 nm) than that of **1**. The full radial π-conjugation along with the two [60]fullerene cages further elevates the $g_{abs}$ value in **4**.

The phosphine-mediated deoxygenative coupling of two open-[60]fullerenes realized the advent of fully-conjugated, double-opened $C_{120}$ nanocarbons with a well-defined structure. The full radial π-conjugation along with the two inherently chiral nanocages allows the molecule to possess the highest dissymmetry factor among known π-conjugated materials. This firmly stimulates the intellectual curiosity for applying them into chiroptoelectronic devices as well as for designing non-naturally occurring inter-[60]fullerene allotropes that are otherwise inaccessible

## Methods
### General

The NMR chemical shifts were reported in ppm with reference to residual protons and carbons of benzene-$d_6$ ($\delta$ 7.15 ppm in ¹H NMR) and ODCB-$d_4$ ($\delta$ 7.20 ppm in ¹H NMR, $\delta$ 132.35 ppm in ¹³C NMR). The ¹H NMR chemical shifts measured for sample solutions in $CS_2$ were reported in ppm with reference to an external standard, i.e., DHO ($\delta$ 4.80 ppm) in a glass sealed capillary inserted inside the NMR tube filled with each sample solution. The ³¹P NMR chemical shifts were reported in ppm with reference to an external standard, i.e., $H_3PO_4$ ($\delta$ 0.00 ppm) in a glass sealed capillary inserted inside the NMR tube filled with $D_2O$.

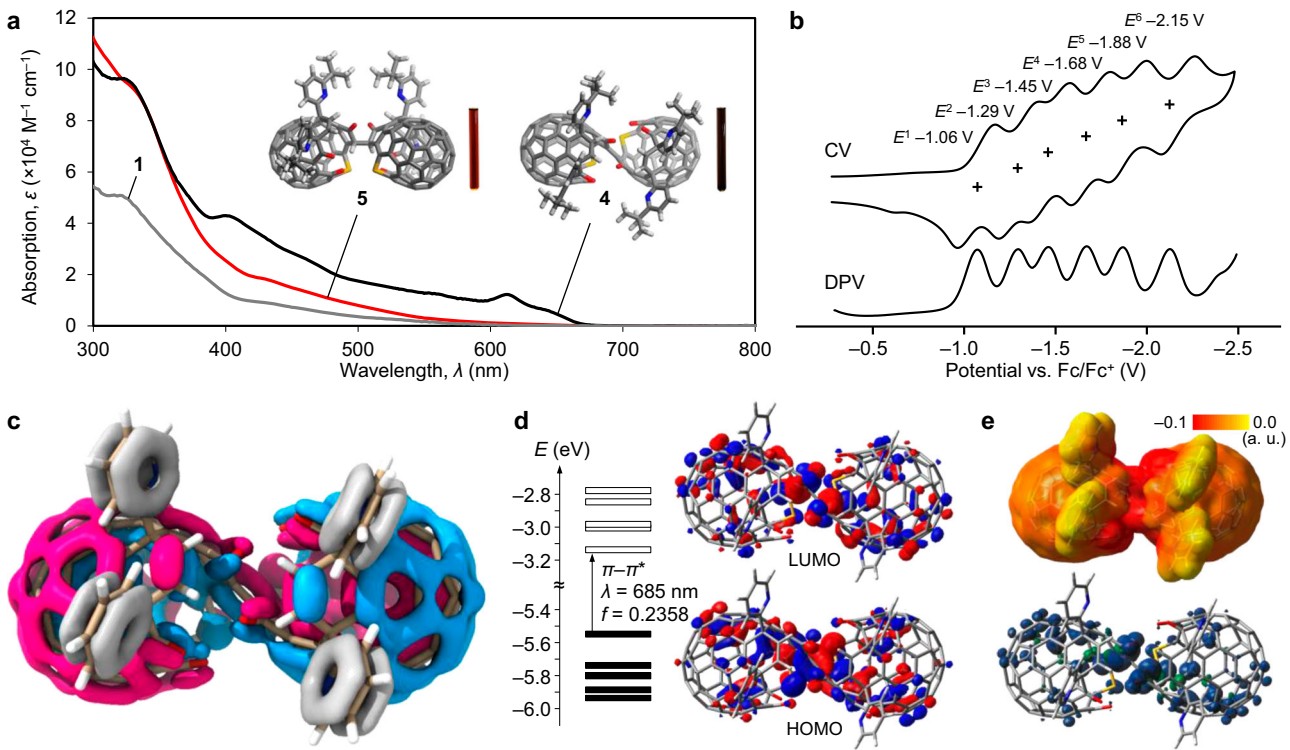

**Fig. 3 | Electronic properties of open-[60]fullerene dimers. a** Absorption spectra of **1**, **4**, and **5** in toluene with selected molecular structures (gray for C, blue for N, red for O, yellow for S, and white for H) and photographs of the solutions. **b** Cyclic and differential pulse voltammograms (CV and DPV) of **4** (Plus signs denote half-wave potentials. 0.5 mM in ODCB, 0.1 M $n$-Bu$_4$N•BF$_4$, 100 mV s$^{-1}$). The y-axis is relative current. **c** LOL-$\pi$ isosurface of **4'** (B3LYP-D3/6-31 G(d,p); color codes, pink/ blue for radially distributed $\pi$-orbitals and gray for $\pi$-orbitals which are not engaged in the caged $\pi$-surface). **d** Optical transitions and molecular orbitals of **4'** (TD CAM-B3LYP-D3/6-31 G(d,p)//B3LYP-D3/6-31 G(d,p), transition energy was scaled by an empirical factor of 72%.). **e** Electrostatic potential (upper) and spin density maps (lower) of (**4'**)$^{-}$ (UB3LYP-D3/6-31 G(d,p)). Source data are provided as a Source Data file.

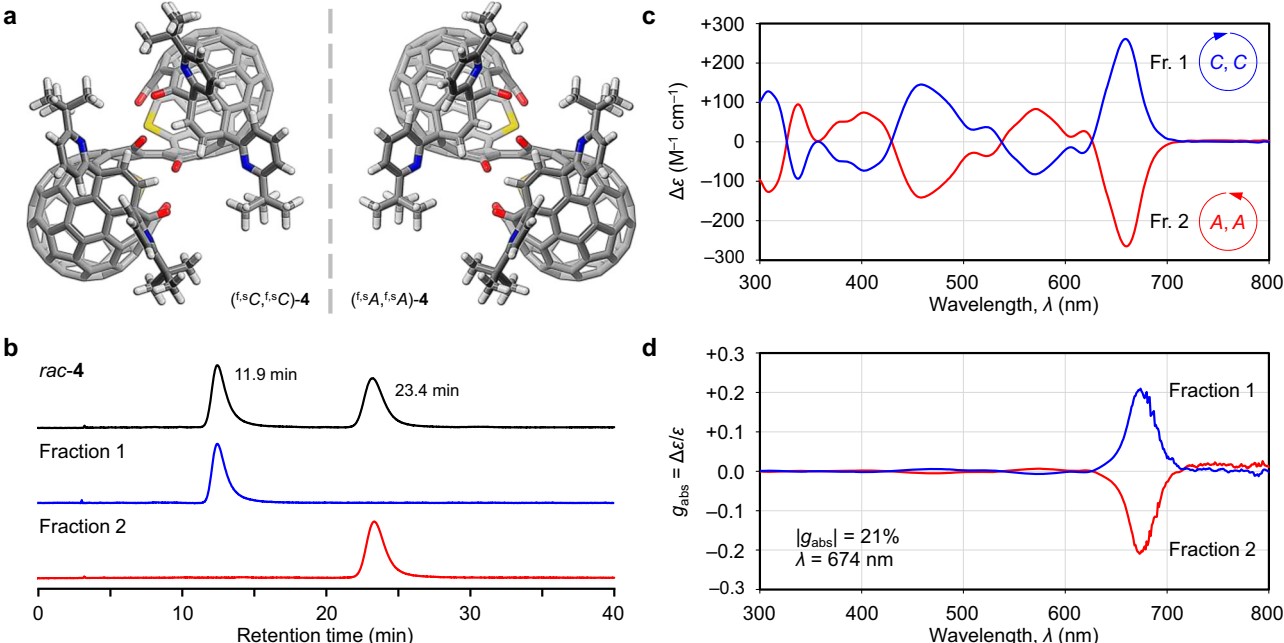

**Fig. 4 | Chiroptical properties of open-[60]fullerene dimer (4). a** Structures of two enantiomers (gray for C, blue for N, red for O, yellow for S, and white for H). **b** Chiral high-performance liquid chromatography charts (toluene, 1 mL min$^{-1}$, 50 °C, 326 nm). **c** Circular dichroism spectra (10 μM in toluene). **d** Dissymmetry factor $g_{abs}$ spectra. Source data are provided as a Source Data file.

APCI (atmospheric pressure chemical ionization) mass spectra were measured on a Bruker micrOTOF-Q II. UV-vis-NIR (ultraviolet-visible-near infrared) absorption spectra were measured with a Shimadzu UV-3150 spectrometer. CD spectra were recorded on a JASCO J720W spectrometer. Fourier transform infrared spectrometer spectra were measured with a Shimadzu IR-Affinity 1 S. Cyclic voltammetry was conducted on a BAS Electrochemical Analyzer ALS620C. The high-performance liquid chromatography (HPLC) was performed with the use of a Cosmosil Buckyprep column (250 mm in length, 4.6 mm in inner diameter) for analytical purpose and the same columns (two directly connected columns; 250 mm in length, 20 mm in inner diameter) for preparative purpose. The chiral HPLC was performed with the use of a CHIRALPAK IF column (250 mm in length, 4.6 mm in inner diameter) for analytical purpose and the same column (250 mm in length, 10 mm in inner diameter) for the preparative purpose. Thin layer chromatography (TLC) was performed on glass plates coated with 0.25 mm thick silica gel 60F-254 (Merck). Column chromatography was performed using PSQ 60B or 100B (Fuji Silysia). All reactions were carried out under Ar atmosphere.

Toluene was purchased from Kanto Chemical Co., Inc. Ethyl acetate (purity: >99.0%) and toluene (purity: >99.0%) were purchased from Nacalai Tesque, Inc. Trimethylphosphine (1.0 M toluene solution) and ODCB (purity: 99%) were purchased from Sigma-Aldrich Co. LLC. Carbon disulfide (purity: >99.0%) was purchased from FUJIFILM Wako Pure Chemical Corporation. Potassium *t*-butoxide (purity: >97.0%) was purchased from Tokyo Chemical Industry Co., Ltd. Unless otherwise noted, materials purchased from commercial suppliers were used without further purification. Compounds **1**−**3** were synthesized according to a literature procedure[21].

## Computational methods

All calculations were conducted with the Gaussian 09 program package. All structures at stationary states were optimized at the (U)B3LYP-D3/6-31 G(d,p) or M06-2X/6-31 G(d,p) level of theory (Supplementary Figs. 29−33 and Supplementary Tables 1−20). All structures were confirmed by the frequency analyses at the same level of theory. Using geometries optimized at the (U)B3LYP-D3/6-31 G(d,p) level of theory, the Kohn-Sham frontier orbitals, spin density maps, and electrostatic potential maps were drawn at the same level of theory. Using geometries optimized at the B3LYP-D3/6-31 G(d,p) level of theory, TD DFT (time-dependent density-functional theory) calculations were conducted at the CAM-B3LYP-D3/6-31 G(d,p) level of theory. The LOL-$\pi$-isosurface was calculated by Multiwfn[26].

## Synthesis

Selected procedures are shown below. The characterization data are described in Supplementary Figs. 2−19 in the Supplementary Information.

## Synthesis of inter-[60]fullerene hybrids

Typical procedure was shown here. Powdery **1** (20.0 mg, 17.6 μmol) was placed into a Schlenk tube and degassed through three vacuum-Ar cycles. ODCB (2.0 mL, 8.8 mM) and then trimethylphosphine (1.0 M in toluene, 35.4 μL, 35 μmol, 2.0 equiv.) were added to the tube. The resulting mixture was heated at 150 °C for 1 h (aluminum block heater). The HPLC chart showed the formation of desired dimers (fig. S1). The chromatographic purification using silica gel (CS₂/AcOEt (20:1) to (5:1)) gave two fractions A and B (14.0 and 3.20 mg, respectively), each of which was a mixture of several compounds. The first eluted fraction A was purified by HPLC equipped with the Buckyprep column which gave **5** (0.74 mg, 0.35 μmol, 4%), **3** (6.53 mg, 5.82 μmol, 33%), and **4** (4.00 mg, 1.79 μmol, 20%) as brown powders. The second eluted fraction B was purified in a similar manner, giving **3** (0.45 mg, 0.40 μmol, 3%) and **6** (1.04 mg, 0.464 μmol, 5%) as brown powders.

## Synthesis of enantiopure inter-[60]fullerene hybrids

Enantiopure $^{f,s}C$-**1** (20.0 mg, 17.6 μmol) was placed into a Schlenk tube and degassed through three vacuum-Ar cycles. ODCB (2.0 mL, 8.8 mM) and then trimethylphosphine (1.0 M in toluene, 35.2 μL, 35 μmol, 2.0 equiv.) were added to the tube. The resulting mixture was heated at 150 °C for 2 h (aluminum block heater). After the reaction, residual PMe₃, its oxide, and ODCB were removed under the reduced pressure. The crude mixture was dissolved in toluene and purified by HPLC (Buckyprep column, 7.5 mL/min, toluene) to give $^{f,s}C$-**2** (2.78 mg, 2.33 μmol, 13%), $^{f,s}A$-**3** (2.85 mg, 2.54 μmol, 14%), a fraction containing ($^{f,s}A,^{f,s}A$)-**5** (ca. 5 mg), and ($^{f,s}A,^{f,s}A$)-**4** (6.31 mg, 2.82 μmol, 32%) as brown powders. The further purification by silica gel column chromatography (toluene) gave ($^{f,s}A,^{f,s}A$)-**5** (2.66 mg, 1.19 μmol, 13%) as a brown powder. In a similar manner, ($^{f,s}C,^{f,s}C$)-**4** and ($^{f,s}C,^{f,s}C$)-**5** were synthesized from $^{f,s}A$-**1**.

## Hydrogenation of 4

Powdery *rac*-**4** (1.54 mg, 0.69 μmol) was placed into a Schlenk tube and degassed through three vacuum-Ar cycles. ODCB (0.50 mL, 0.14 mM) and then trimethylphosphine (1.0 M in toluene, 1.6 μL, 1.6 μmol, 2.3 equiv.) were added to the tube. The resulting mixture was heated at 150 °C for 24 h (aluminum block heater). After the reaction, residual PMe₃, its oxide, and ODCB were removed under the reduced pressure. The chromatographic purification using silica gel (CS₂/toluene (1:1) to toluene) gave *rac*-**5** (0.75 mg, 0.33 μmol) in 49% isolated yield as a brown powder.

## Crystallography

Single crystals of **4** and **5** were obtained from CS₂/CHCl₃ and CS₂/toluene solutions, respectively (Supplementary Figs. 27, 28). Intensity data were collected at 100 K. The structure was solved by direct methods (SHELXT-2014/5) and refined by the full-matrix least-squares on $F^2$ (SHELXL-2018/3)[37].

## Association constant

In benzene-$d_6$, *rac*-**5** showed two sets of ¹H signals corresponding to **5** and (**5**)₂ owing to the association occurring slower than the NMR time scale (Supplementary Fig. 20) whereas ($^{f,s}A,^{f,s}A$)-**5** showed only single component under the same conditions (Supplementary Fig. 21). The association did not cause significant change in a chemical shift of the encapsulated H₂O molecule. Since two proton signals corresponding to the pyridyl groups for *rac*-**5** showed remarkable downfield shifts, the association might be promoted by a heterochiral recognition. Such association behavior was not confirmed for *rac*-**4**. By integrating the peak area, the association constant of **5** was determined to be $1.11 \times 10^4$ M⁻¹ in benzene-$d_6$ at 300 K ($\Delta G = -1.43$ kcal mol⁻¹). The diffusion coefficients were determined to be $D = 4.40 \times 10^{-10}$ (**5**) and $3.71 \times 10^{-10}$ ((**5**)₂) m² s⁻¹ using 1.5 mM solutions in benzene-$d_6$ (800 MHz, 3.0-mmφ NMR tube).

## Electrochemical analysis

Cyclic voltammetry was conducted using a three-electrode cell with a glassy carbon working electrode, a platinum wire counter electrode, and an Ag/AgNO₃ reference electrode. The measurements were carried out under N₂ atmosphere. The concentrations of sample solutions were set to 0.50 mM in ODCB except for **1** (1.0 mM). As a supporting electrolyte, *n*-Bu₄N•BF₄ (0.10 M) was used. The scan rate was set to 100 mV s⁻¹. The redox potentials were calibrated with ferrocene used as an internal standard which was added after each measurement (Supplementary Fig. 13).

## UV-vis-NIR absorption spectroscopy

The measurements were conducted using racemic samples except for **5** (Supplementary Figs. 14−16) since *rac*-**5** was hardly dissolved in organic solvents due to the self-association behavior while the enantiomerically pure sample, ($^{f,s}C,^{f,s}C$)-**5**, could be readily dissolved in common organic solvent.

**Reporting summary**

Further information on research design is available in the Nature Portfolio Reporting Summary linked to this article.

## Data availability

Crystallographic data for the structures reported in this Article have been deposited at the Cambridge Crystallographic Data Center, under deposition numbers 2211312 (**4**) and 2211311 (**5**). Copies of the data can be obtained free of charge via http://www.ccdc.cam.ac.uk/structures/. The data that support the findings of this study are available from the corresponding authors upon request. Source data are provided with this paper.

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

## Acknowledgements

This work was supported by the JSPS KAKENHI Grant Number JP23H01784 (Y.M.) and JP22H04538 (Y.H.), The Mazda Foundation (Y.H.), and Advanced Technology Institute Research Grants 2023 (Y.H.). The NMR measurements were partly supported by the Joint Usage/Research Center (JURC) at the ICR, Kyoto University. We are grateful to Assist. Prof. Yoshihiro Ueda and Prof. Takeo Kawabata (Kyoto university) for their support of CD measurements.

## Author contributions

Y.H. and Y.M. launched the project. Y.H. conceived the design. Y.H. and S.O. conducted all experiments. Y.H. performed theoretical calculations

and crystallographic analyses. Y.H. wrote the manuscript and discussed the results with all authors.

## Competing interests

The authors declare no competing interests.
