## [Peer Review File · Nature Communications]

Synthesis of inter-[60]fullerene conjugates with inherent chiralityREVIEWER COMMENTS

Reviewer #1 (Remarks to the Author):

C60, as the most abundant molecular carbon cluster, remains in the spotlight due to its remarkable topological and physical properties. Moreover, the recent coalescence of two or more fullerene cages has emerged as a highly significant topic in the field of material science. In this comprehensive research article, the authors have successfully created covalent assemblies of inherently chiral open-C60 through organic reactions. Furthermore, the structures and the chiroptical properties of the dimer compounds were well characterized by single crystal. Finally, I recommend that this paper be considered for publication in Nature Communications after revisions.

1. I think it is necessary to explore some application of this material to satisfy the journal as a comprehensive journal, in order to attract broader interest.

2. One of the crystal structure files contains numerous level B errors, which should be minimized to the greatest extent possible.

3. In Figure 1(c), the synthesis routes should be adjusted to enhance clarity. For instance, the distinct open-[60]fullerenes could be labeled with the code names 1, 2, and 3.

4. While it was noted that the dissymmetry factor of 4 is one of the largest ever reported for conventional chiral organic molecules, it would be beneficial to provide references to one or two molecules with similarly substantial dissymmetry factors.

5. The author needs to list the dissymmetry factor materials before and compare them with this work.

6. The synthesis procedures of open-C60 monomer 1, 2, and 3 should be added in the Supplementary Information file.

Reviewer #2 (Remarks to the Author):

The authors report on a highly fascinating compound, namely, a two open bis-[60] fullerene opening the new compound class of double-opened C120 nanocarbons with a well-defined structure. They were able to fully characterize this molecule including x-ray crystallography, and electrochemical investigations. They were also able to separate enantiopure isomers and carry out chiroptical investigation. Importantly, these stereoisomers exhibit excellent chiroptical activities with an exceptionally large dissymmetry factor of 0.21 breaking the record for known chiral organic molecules. This work is not only an excellent piece of fullerene- and nanochemistry but also an important contribution to fundamental organic chemistry in general. Publication in Nature Commun. is recommended as it stands

Reviewer #3 (Remarks to the Author):

The work "A Synthetic Inter[60]fullerene Conjugate with Inherent Chirality", by Hashikawa, Okamoto and Murata describes the synthesis, characterization, electronic and chiroptical properties of novel open cage fullerene dimers. The dimers were prepared from the corresponding monomers, which are known compounds. The structural characterization (IR, NMR, X-ray, HRMS) of the dimers is presented in detail. Among them, a homo-chiral dimer, where the two fullerene spheres are connected via a C=C double bond stands out. This conjugated dimer shows distinguished electronic (absorption spectra, cyclic voltammetry, DFT), as well as chiroptical properties (chiral HPLC, CD), as compared to the non-conjugated ones (spheres connected via a C-C single bond). Especially regarding the chiroptical properties, the conjugated dimer shows an exceptionally large value for absorption dissymmetry factor $g_{abs}=0.21$, characterized as among the largest ever reported among organic molecules, attributed to "The full radial π -conjugation along with the two inherently chiral nanocages allows the molecule to

break the record of dissymmetry factors." This value is the highlight of the present work. Finally, due to the large orifices of the fullerene spheres, small molecules (Ar, N₂, H₂O) may be encapsulated inside the spheres.

The work is well presented, and the claims are thoroughly examined. Although the gabs=0.21 is indeed impressive, the same group has recently presented an open cage fullerene monomer, with gabs=0.20 [ref. 30 in the text, "Chiral Open-[60]Fullerene Ligands with Giant Dissymmetry Factors" J. Am. Chem. Soc. 2022, 144, 18829–18833]. The almost similar gabs values for a monomer and a conjugated dimer, suggest a rather more complex explanation for the observed value, to the one presented above by the authors.

I recommend publication of the present article, as it presents a sophisticated synthesis, of a record gabs molecule, on a rapidly expanding topic. However, I would suggest:

1. The fullerene monomer with gabs=0.20 (2022JACS article mentioned above) should be also mentioned in this work for comparison, and suggest (if any) possible explanations for the marginally larger value of the present work dimer molecule.

2. A minor suggestion. In Fig. 1, the inter-sphere bonds are pointed by arrows as "sp³" and "sp²". These characterize atoms, not bonds. It should be more appropriate a "σ-bond"/"π-bond", single/double bond or sp³-sp³/sp²-sp².

Reviewer #4 (Remarks to the Author):

This is a very beautiful piece of chemistry. Synthetically the work follows the usual approach of the Murata lab, but the outcome is really interesting, as a potentially first example of a concave-convex conjugation of such complexity. In this context, the authors should discuss known buckybowl dimers, as potential reference systems for the present case. The work is nicely illustrated and contains a sufficient amount of data to substantiate the key claims.

While I like the chemistry very much, I am really upset by the low quality of writing that is simply unacceptable in any scientific contribution, let alone in a top chemistry journal. Some examples are mentioned below, but the text is ridden with problems, ranging from misused prepositions to unnecessarily boastful statements. I do hope the authors can fix these issues properly in the revision, because I would really like to see their work in print in Nat. Comm.

Specific comments

Fig. 1 caption. Quotation marks not needed for "Chiral".

Fig. 2a. The combination of the two syntheses (racemic and enantiopure) leads to considerable confusion, because the two cases are not clearly separated. Two arrows emerge from C-1 (in spite of the Authors' attempt to differentiate the two routes). A and C enantiomers are color-coded (gray and blue, respectively), but then we get the homodimers that are gray colored, and labeled as either rac or (A,A) (i.e. opposite to the starting C enantiomer). The figure should be split, e.g. into (a1) labeled, structures, (b2) rac reactions (using labels only), and (b3) enantiopure reactions (again using labels only). The cause of of stereodescriptor inversion in the dimers should be explained.

"constituted of"
grammar

"With the aid of its singularity in physical nature such as electron-accepting², superconducting^{3,4}, and ferromagnetic behavior⁵, the esthetics originated from the congenital topology has continued to touch the hearts of scientists since the early 1990s, during which a number of hypothetical roundish polyhedrons have been proposed by a recombination of atomic arrangement in C₆₀ and/or fusion of multiple molecules of C₆₀."

This is a representative example of multiple stylistic and grammatical problems the manuscript

(singularity in physical nature, originated from, congenital, touch the hearts, roundish, recombination). The authors should fix their grammar, avoid risky metaphors and word choices, and ask an experienced native-speaker to proof-read the text.

The issues are too numerous to list here, however, it is absolutely necessary for the authors to revise the language before the manuscript is considered for acceptance.

"hybridization of the inner and outer pz lobes"

this is not hybridization

Witting

"transient generation of a 1,3-betaine"

"formed in 4 and 5%"

...yield

"are driven by a thermodynamic control"

"Thermodynamic control" normally means that the process is reversible. Here the hydrogenation is clearly non-reversible (there is also a deoxygenation step that is not mentioned in the manuscript). For the initial addition-elimination, it is not clear if the mixture can equilibrate because it will depend on the relative rates of addition-elimination and deoxygenation. The authors need to revise the description and potentially include TS calculations in Supp. Fig. 29.

"The seeds inside the cages were refined to be N₂ and Ar with a superiority for the former. Within the crystal, (N₂:Ar)₄, simultaneously accommodating a different seed in each cage, accounts for a third."

This statement is unclear (seeds? superiority?) Apparently the authors describe disorder in the crystal, which should be done using established crystallographic terminology.

"non-conjugated dimer 5 is assembled in a form of dimer by a heterochiral recognition where the stereogenic centers, i.e., the four orifices, are bound to the pyridyl groups via hydrogen-bondings in a complementary manner (Fig. 2c)."

This needs to be revised, it is initially not clear that a dimer of dimers is observed.

The H bond donors should be specified in the discussion (are they really stereogenic as written?). It is not easy to see this clearly in the figure (which could also be improved).

"In benzene-d₆, rac-5 showed two sets of 1H signals corresponding to 5 and (5)₂ owing to the association occurring slower than the NMR time scale"

"demonstrative of the event being thermodynamically favorable"

this is redundant: if it happens under equilibrium, it must be thermodynamically favorable.

"Upon seeing"

subject disagreement

"(calibration factor of 0.72 28)"

ref 28 does not discuss this aspect. The value used there is 0.75. It is not clear what the authors do with the spectrum (what does "calibration" mean, in the first place?). The systematic error of TD predictions is typically expressed in eV relative to the energy gap. The authors should describe their treatment in detail, and provide a computational reference that justifies their approach.

Fig. 3c Physical significance of the color coding should be explained (purple vs. blue vs. gray).

"the gabs value of 4 is abnormally large on the ground of the effective radial n-conjugation along with the two [60]fullerene cages"

This is a key statement in the paper, but it must be supported with some justification.

“coalescence”

formation of one bond is hardly a “coalescence”

“fullerites”

“We anticipate that the science of fullerenes is now reborn to again stand in the spotlight so that we could synthesize and handle non-naturally occurring inter[60]fullerene allotropes that are otherwise inaccessible.”

This is a highly inappropriate final sentence.

Point-by-Point Responses

Title: A Synthetic Inter[60]fullerene Conjugate with Inherent Chirality

Authors: Yoshifumi Hashikawa, Shu Okamoto, and Yasujiro Murata*

Revised for publication in *Nat. Commun.* on 2023/11/22

To Reviewer 1

We express our sincere gratitude to the Reviewer 1 for the fruitful comments. We have revised our manuscript based on comments from the Reviewer 1.

Reviewer 1's Comments

C60, as the most abundant molecular carbon cluster, remains in the spotlight due to its remarkable topological and physical properties. Moreover, the recent coalescence of two or more fullerene cages has emerged as a highly significant topic in the field of material science. In this comprehensive research article, the authors have successfully created covalent assemblies of inherently chiral open-C60 through organic reactions. Furthermore, the structures and the chiroptical properties of the dimer compounds were well characterized by single crystal. Finally, I recommend that this paper be considered for publication in Nature Communications after revisions.

1. I think it is necessary to explore some application of this material to satisfy the journal as a comprehensive journal, in order to attract broader interest.

Our Response #1: Thank you for important suggestion. Currently, we are planning to survey the potential application of these materials such as enantiopure organic semiconductors and CPL detectors. In this manuscript, we have shown the intense broadband absorption and excellent chiroptical activity of the conjugated dimer. Therefore, we believe that these properties themselves attract broad interest from readers of this journal. In the near future, we will perform further study on application of these materials into optical and electronic devices and report the results separately.

2. One of the crystal structure files contains numerous level B errors, which should be minimized to the greatest extent possible.

Our Response #2:

We have revised the crystal structure file. As pointed out by the Reviewer 1, the crystal structure file of compound **5** includes 21 B-level alerts. 13 B-level alerts were mainly attributed to a number of disordered solvent molecules such as CS₂ and toluene although they were solved using appropriate disordered models with 11 (for CS₂) and 14 (for toluene) dispositions to minimize the alerts as much as possible. And, these alerts are inevitable because many solvent molecules are filled in the voids generated between the fullerene cages. In the revised cif file, we added comments toward these B-level alerts though 2 B-level alerts have been solved appropriately.

Remaining 8 alerts (D-H Without Acceptor) are caused by the four H₂O molecules encapsulated inside the fullerene cages. Since encapsulated H₂O molecules are isolated from the ambient environment, they have no hydrogen-bonding acceptors. Thus, there

are indeed no structural issues on the H₂O molecules. Thus, in the revised cif file, we added comments toward these B-level alerts.

[Revised] The cif file (2sp3PP_a.cif) was revised and re-deposited to CCDC with keeping the original registration number (CCDC 2211311). The detailed comments are found in the revised cif file.

3. In Figure 1(c), the synthesis routes should be adjusted to enhance clarity. For instance, the distinct open-[60]fullerenes could be labeled with the code names 1, 2, and 3.

Our Response #3: According to the suggestion from the Reviewer 1, we have revised Figure 1c.

[Revised, Figure 1c] The labels of the compounds were shown below each structure.

4. While it was noted that the dissymmetry factor of 4 is one of the largest ever reported for conventional chiral organic molecules, it would be beneficial to provide references to one or two molecules with similarly substantial dissymmetry factors.

Our Response #4: In the original manuscript, we have already provided two references (Refs. 33 and 34) for materials showing similarly large dissymmetry factors. For your information, we show them here (In the revised manuscript, the reference numbers were changed to Refs. 35 and 36).

35. Schippers, P. H. & Dekkers, H. P. J. M. Circular Polarization of Luminescence as a Probe for Intramolecular ¹n π^* Energy Transfer in *meso*-Diketones. *J. Am. Chem. Soc.* **105**, 145–146 (1983).

36. Sato, S., Yoshii, A., Takahashi, S., Furumi, S., Takeuchi, M. & Isobe, H. Chiral intertwined spirals and magnetic transition dipole moments dictated by cylinder helicity. *Proc. Natl. Acad. Sci. U. S. A.* **114**, 13097–13101 (2017).

5. The author needs to list the dissymmetry factor materials before and compare them with this work.

Our Response #5: According to the Reviewer 1's comment, we have added a brief comment on the two previous examples.

[**Added, In the section of Chiroptical Properties**] “This is, to the best of our knowledge, among the largest ever reported for conventional chiral organic molecules as their g_{abs} values are in general found at an order of 10^{-4} – 10^{-3} ^{32–34}. **The similarly large values have been reported for hexahydropentalenone (ca. 0.2 supposedly at UV region)³⁵ and cycloarylene (0.167 at 443 nm)³⁶.**”

6. The synthesis procedures of open-C60 monomer 1, 2, and 3 should be added in the Supplementary Information file.

Our Response #6: The synthetic procedures of **1–3** were shown in the previous report. To avoid duplicated description, we have added a brief comment in the revised Supplementary Information.

[**Added, Supplementary Information, 1. General**] “**Compounds 1, 2, and 3 were synthesized according to literature procedures (see details in Ref. 21 listed in manuscript).**”

To Reviewer 2

We appreciate to the high evaluation and comments from the Reviewer 2. We have revised our manuscript based on other three Reviewers. Please find the responses to the Reviewers 1, 3, and 4.

Reviewer 2's Comments

The authors report on a highly fascinating compound, namely, a two open bis-[60] fullerene opening the new compound class of double-opened C₁₂₀ nanocarbons with a well-defined structure. They were able to fully characterize this molecule including x-ray crystallography, and electrochemical investigations. They were also able to separate enantiopure isomers and carry out chiroptical investigation. Importantly, these stereoisomers exhibit excellent chiroptical activities with an exceptionally large dissymmetry factor of 0.21 breaking the record for known chiral organic molecules. This work is not only an excellent piece of fullerene- and nanochemistry but also an important contribution to fundamental organic chemistry in general. Publication in Nature Commun. is recommended as it stands

To Reviewer 3

We are grateful to the evaluation and comments from the Reviewer 3. We carefully considered the suggestions from the Reviewer 3 and responses were shown below.

Reviewer 3's Comments

The work "A Synthetic Inter[60]fullerene Conjugate with Inherent Chirality", by Hashikawa, Okamoto and Murata describes the synthesis, characterization, electronic and chiroptical properties of novel open cage fullerene dimers. The dimers were prepared from the corresponding monomers, which are known compounds. The structural characterization (IR, NMR, X-ray, HRMS) of the dimers is presented in detail. Among them, a homo-chiral dimer, where the two fullerene spheres are connected via a C=C double bond stands out. This conjugated dimer shows distinguished electronic (absorption spectra, cyclic voltammetry, DFT), as well as chiroptical properties (chiral HPLC, CD), as compared to the non-conjugated ones (spheres connected via a C-C single bond). Especially regarding the chiroptical properties, the conjugated dimer shows an exceptionally large value for absorption dissymmetry factor $g_{abs}=0.21$, characterized as among the largest ever reported among organic molecules, attributed to "The full radial π -conjugation along with the two inherently chiral nanocages allows the molecule to break the record of dissymmetry factors." This value is the highlight of the present work. Finally, due to the large orifices of the fullerene spheres, small molecules (Ar, N₂, H₂O) may be encapsulated inside the spheres.

The work is well presented, and the claims are thoroughly examined. Although the $g_{abs}=0.21$ is indeed impressive, the same group has recently presented an open cage fullerene monomer, with $g_{abs}=0.20$ [ref. 30 in the text, "Chiral Open-[60]Fullerene Ligands with Giant Dissymmetry Factors" J. Am. Chem. Soc. 2022, 144, 18829–18833]. The almost similar g_{abs} values for a monomer and a conjugated dimer, suggest a rather more complex explanation for the observed value, to the one presented above by the authors.

I recommend publication of the present article, as it presents a sophisticated synthesis, of a record g_{abs} molecule, on a rapidly expanding topic. However, I would suggest:

1. The fullerene monomer with $g_{abs}=0.20$ (2022JACS article mentioned above) should be also mentioned in this work for comparison, and suggest (if any) possible explanations for the marginally larger value of the present work dimer molecule.

Our Response #1: In the revised manuscript, we have added the information of our previous example for comparison with those presented herein. We also added brief discussion on the magnitude of dissymmetry factors.

[Added, In the Section of **Chiroptical Properties**] "**Monomer 1** showed $g_{abs} = \pm 0.063$ at 648 nm, which is rather smaller than another open-[60]fullerene reported previously (± 0.20 at 710 nm)³⁰. This might arise from the large 17-atom-ring orifice in **1** causing a flexibility of the caged structure, which partly breaks radial π -conjugation in solution while the latter is structurally rigid owing to the small 12-atom ring. As a result of an enhanced rigidity by dimerization, **5** attains a twice larger g_{abs} value (± 0.063 at 648 nm) than that of **1**. ~~Even when compared with monomer **1** ($g_{abs} = \pm 0.036$ at 652 nm) and non-conjugated dimer **5** ($g_{abs} = \pm 0.063$ at 648 nm), the g_{abs} value of **4** is abnormally large on the ground of the effective radial π conjugation along with the two [60]fullerene cages.~~"

2. A minor suggestion. In Fig. 1, the inter-sphere bonds are pointed by arrows as “sp3” and “sp2”. These characterize atoms, not bonds. It should be more appropriate a “σ-bond”/”π-bond”, single/double bond or sp3-sp3/sp2-sp2.

Our Response #2: According to the comment, we have revised Figure 1 as shown below.

[Revised, Figure 1] The labels were changed to “sp³-sp³” and “sp²-sp²”.

To Reviewer 4

We appreciate careful reviewing by the Reviewer 4. According to the comments from the Reviewer 4, we have revised our manuscript. The responses were listed below.

Reviewer 4's Comments

This is a very beautiful piece of chemistry. Synthetically the work follows the usual approach of the Murata lab, but the outcome is really interesting, as a potentially first example of a concave-convex conjugation of such complexity. In this context, the authors should discuss known buckybowl dimers, as potential reference systems for the present case. The work is nicely illustrated and contains a sufficient amount of data to substantiate the key claims.

Our Response #1: We added a brief discussion of the Buckybowl dimers as suggested by the Reviewer 4.

[Added, In the Section of **Electronic Properties**] “This concave–convex conjugation is characteristic to the open-[60]fullerene conjugate whereas it could not be discriminated from concave–concave/convex–convex conjugations for analogous Buckybowl dimers consisting of corannulene²⁹ or sumanene³⁰ due to possible bowl inversion as well as rotation even along the olefin linkage.”

[Added, References]

29. Eisenberg, D., Filatov, A. S., Jackson, E. A., Rabinovitz, M., Petrukhina, M. A., Scott, L. T. & Shenhar, R. Bicolorannulene: Stereochemistry of a C₄₀H₁₈ Biaryl Composed of Two Chiral Bowls. *J. Org. Chem.* **73**, 6073–6078 (2008).
30. Amaya, T., Ito, T. & Hirao, T. Synthesis and Characterization of Bisumanenylidene. *Eur. J. Org. Chem.* 3531–3535 (2014).

While I like the chemistry very much, I am really upset by the low quality of writing that is simply unacceptable in any scientific contribution, let alone in a top chemistry journal. Some examples are mentioned below, but the text is riddled with problems, ranging from misused prepositions to unnecessarily boastful statements. I do hope the authors can fix these issues properly in the revision, because I would really like to see their work in print in Nat. Comm. Specific comments

Fig. 1 caption. Quotation marks not needed for “Chiral”.

Our Response #2: In the revised manuscript, we have removed quotation marks from Figure 1.

[Revised, Figure 1]

Fig. 2a. The combination of the two syntheses (racemic and enantiopure) leads to considerable confusion, because the two cases are not clearly separated. Two arrows emerge from C-1 (in spite of the Authors' attempt to differentiate the two routes). A and C enantiomers are color-coded (gray and blue, respectively), but then we get the homodimers that are gray colored, and labeled as either rac or (A,A) (i.e. opposite to the starting C enantiomer). The figure should be split, e.g. into (a1) labeled, structures, (b2) rac reactions (using labels only), and (b3) enantiopure reactions (again using labels only). The cause of stereodescriptor inversion in the dimers should be explained.

Our Response #3: We have revised Figure 2a as shown below. We believe the revised figure does not cause misleading. The comment on the stereodescriptor change was included in the revised manuscript. The change in stereodescriptor from C (monomer) to A (dimer) does not mean chiral inversion. This is due to the change in priority order of C1 and C2 by the loss of oxygen atoms from the orifices in dimers.

[Revised, Figure 2a]

[Added, In the section of **Synthesis and Structure**] “Note that the change in stereodescriptor does not indicate a chiral inversion but is because of the reversed priority order of C1 and C2 in both **4** and **5** by losing oxygen atoms.”

“constituted of”

Our Response #4: We have revised the statement.

[Revised, Abstract] “... during an oligomerization process. Herein, we showcase the rational synthesis of covalent assemblies ~~consisting constituted~~ of inherently chiral open-[60]fullerenes. The crystallographic analyses ...”

grammar

“With the aid of its singularity in physical nature such as electron-accepting², superconducting^{3,4}, and ferromagnetic behavior⁵, the esthetics originated from the congenital topology has continued to touch the hearts of scientists since the early 1990s, during which a number of hypothetical roundish polyhedrons have been proposed by a recombination of atomic arrangement in C₆₀ and/or fusion of multiple molecules of C₆₀.”

This is a representative example of multiple stylistic and grammatical problems the manuscript (singularity in physical nature, originated from, congenital, touch the hearts, roundish, recombination). The authors should fix their grammar, avoid risky metaphors and word choices, and ask an experienced native-speaker to proof-read the text.

The issues are too numerous to list here, however, it is absolutely necessary for the authors to revise the language before the manuscript is considered for acceptance.

Our Response #5: We have revised the description.

[Revised, Introduction] ~~“With the aid of its singularity in physical nature such as electron-accepting², superconducting^{3,4}, and ferromagnetic behavior⁵, the esthetics originated from the congenital topology has continued to touch the hearts of scientists since the early 1990s, during which a number of hypothetical roundish polyhedrons have been proposed by a recombination of atomic arrangement in C₆₀ and/or fusion of multiple molecules of C₆₀.”~~ Inspired by its characteristic physical nature such as electron-accepting², superconducting^{3,4}, and ferromagnetic behavior⁵, a variety of hypothetical carbon-based polyhedrons have been proposed, since the early 1990s, by rearranging atomic coordinates of C₆₀ and/or fusing multiple molecules of C₆₀.”

“hybridization of the inner and outer p_z lobes”
this is not hybridization

Our Response #6: We have revised the corresponding sentence.

[Revised, Introduction] “Herein, we showcase the synthesis and solid-state structure of the first conjugated inter[60]fullerene hybrid. By the use of an open-[60]fullerene^{18,19} as a pairing molecule (Fig. 1b), the full π -conjugation would be realized along with the two cages in a radial manner which enables an effective ~~conjugation hybridization~~ of the inner and outer p_z lobes (Fig. 1c).”

Witting

Our Response #7: We have revised the typo.

[Revised, Introduction] “..., i.e., ~~Wittig~~ ~~Witting~~ reaction, aldol condensation, and ...”

“transient generation of a 1,3-betaine”

Our Response #8: We have revised the corresponding sentence.

[Revised, In the section of **Synthesis and Structures**] “Once the third strategy, that is a phosphine-mediated sequential deoxygenation, was adopted, conjugated dimer **4** was obtained in 20% isolated yield (Fig. 2a), ~~in which 1-phosponium-3-oxobetaine²¹ might be generated as a key intermediate being suggestive of a transient generation of a 1,3-betaine intermediate~~ by the reaction of **1** with the phosphine (Fig. 1c)²¹.”

“formed in 4 and 5%”

...yield

Our Response #9: We have revised all corresponding parts.

[Revised, In the section of **Synthesis and Structures**] “In the same reaction, hydrogenated dimers, **5** and **6**, were also formed in 4 and 5% ~~yields~~, respectively, in which the hydrogenation is assisted by trimethylphosphine^{22,23} so that the conversion of **4** into **5** took place in 49% ~~yield~~.”

“are driven by a thermodynamic control”

“Thermodynamic control” normally means that the process is reversible. Here the hydrogenation is clearly non-reversible (there is also a deoxygenation step that is not mentioned in the manuscript). For the initial addition-elimination, it is not clear if the mixture can equilibrate because it will depend on the relative rates of addition-elimination and deoxygenation. The authors need to revise the description and potentially include TS calculations in Supp. Fig. 29.

Our Response #10: We have revised the corresponding description. As pointed by the Reviewer 4, TS calculations are informative to deeply understand the reaction mechanism. Since we would like to pay more attention to the structures and properties rather than reaction mechanism, we decided to simplify the comments on calculations as follows.

[Revised, In the section of **Synthesis and Structures**] “According to theoretical calculations (Supplementary Fig. 29), **6** is more thermodynamically stable than **5** ~~the dimerization and subsequent hydrogenation are driven by a thermodynamic control~~.”

“The seeds inside the cages were refined to be N₂ and Ar with a superiority for the former. Within the crystal, (N₂:Ar)₄, simultaneously accommodating a different seed in each cage, accounts for a third.”

This statement is unclear (seeds? superiority?) Apparently the authors describe disorder in the crystal, which should be done using established crystallographic terminology.

Our Response #11: We have revised the corresponding description.

[Revised, In the section of **Synthesis and Structures**] “The **chemical species found seeds** inside the cages were refined **to be as a disorder of N₂ and Ar** ~~with a superiority for the former. Within the crystal, (N₂:Ar)₄, simultaneously accommodating a different chemical species seed in each cage, accounts for a third.~~”

“non-conjugated dimer 5 is assembled in a form of dimer by a heterochiral recognition where the stereogenic centers, i.e., the four orifices, are bound to the pyridyl groups via hydrogen-bondings in a complementary manner (Fig. 2c).”

This needs to be revised, it is initially not clear that a dimer of dimers is observed.

The H bond donors should be specified in the discussion (are they really stereogenic as written?). It is not easy to see this clearly in the figure (which could also be improved).

Our Response #12: The structural information about dimer of dimer was carefully described in the revised manuscript. The hydrogen-bondings were formed between pyridyl protons (H donor) and carbonyl groups (H acceptor). The location of the CH...O type hydrogen-bondings was clearly indicated in the revised figure where the bond lengths were also shown. For fullerene chirality, atomic arrangement is the chiral origin as we had stated in the original manuscript. This point was described more clearly in the revised manuscript.

[Revised, In the section of **Synthesis and Structure**] “**Within an asymmetric unit, two crystallographically independent molecules were found for non-conjugated dimer 5**, in which the two molecules, i.e., (^{f,s}C,^{f,s}C)-5 and (^{f,s}A,^{f,s}A)-5, are assembled as a dimer so that four carbon cages are arranged in a tetrahedral manner (Fig. 2c). This heterochiral recognition is enabled by multiple hydrogen-bondings complementarily formed between pyridyl protons and carbonyl oxygens on the four cages whose atomic arrangement is a stereogenic element. ~~Curiously enough, non-conjugated dimer 5 is assembled in a form of dimer by a heterochiral recognition where the stereogenic centers, i.e., the four orifices, are bound to the pyridyl groups via hydrogen bondings in a complementary manner (Fig. 2c).~~”

[Revised, Figures 2b and c] The Figures 2b and 2c were revised as shown below. The hydrogen-bondings were indicated by yellow dot lines. The distances were also shown. We believe that the revised figure is enough clear to see the association modes.

“In benzene-*d*₆, *rac*-5 showed two sets of 1H signals corresponding to 5 and (5)₂ owing to the association occurring slower than the NMR time scale”

Our Response #13: We have revised the corresponding description.

[Revised, In the section of **Synthesis and Structures**] “The ¹H NMR spectrum of **In benzene-*d*₆, *rac*-5** in benzene-*d*₆ showed signals of both ~~two sets of ¹H signals corresponding to 5 and (5)₂~~ owing to the ~~slow association occurring slower than the NMR time scale~~ whereas $(f,sA,f,sA)-5$ showed only single component under the same conditions.”

“demonstrative of the event being thermodynamically favorable”
this is redundant: if it happens under equilibrium, it must be thermodynamically favorable.

Our Response #14: We have revised the corresponding description.

[Revised, In the section of **Synthesis and Structures**] “The association constant was determined to be $1.11 \times 10^4 \text{ M}^{-1}$ in benzene-*d*₆ at 300 K, ~~which corresponds to demonstrative of the event being thermodynamically favorable ($\Delta G = -1.43 \text{ kcal/mol}$).~~”

“Upon seeing”
subject disagreement

Our Response #15: We have revised the corresponding description.

[Revised, In the section of **Electronic Properties**] “~~The Upon seeing the~~ localized orbital locator (LOL) isosurface (Fig. 3c);^{26,27} ~~confirms the π -orbitals are~~ delocalized

over the entire carbon skeleton through the olefin linkage, ~~where confirming~~ a double π -surface ~~is~~ arranged in a radial manner.”

“(calibration factor of 0.72 28)”

ref 28 does not discuss this aspect. The value used there is 0.75. It is not clear what the authors do with the spectrum (what does “calibration” mean, in the first place?). The systematic error of TD predictions is typically expressed in eV relative to the energy gap. The authors should describe their treatment in detail, and provide a computational reference that justifies their approach.

Our Response #16: It is well-known that the optical transition energies obtained by TD DFT calculations are usually overestimated when compared with experimental values, especially for large molecular systems over ca. 100 atoms as well as in the case where charge-transfer transitions are involved. In addition, it is also known that there is a linear correlation between the calculated optical transition energies and experimentally observed λ_{max} . To reduce the calculations cost at the very high level, the use of simple scaling factor is beneficial to reasonably predict the transition energies. Therefore, scaling factors are often used for discussing the absorption spectra of π -conjugated materials. To this end, we previously found that the empirical correction by a scaling factor of 0.72 or 0.75 for the transition energies well-reproduces experimental values. In this manuscript, the overestimated transition energies were scaled by 72%. This corresponds to the correction of λ by $1/0.72$.

For Ref. 28 in the original manuscript, we had cited our previous report which uses a scaling factor of 0.75. To avoid any misleading and to show the validity of the use of a scaling factor, in the revised manuscript, we replaced this reference with a computational reference which discusses the linear correlation between the calculated optical transition energies and experimentally observed λ_{max} . In addition, at the end of the corresponding description, we put another reference (Ref. 21) where we had previously applied a scaling factor of 0.72.

We also gave a brief comment on what we treated for the transition energies in the revised manuscript.

[Revised, In the section of **Electronic Properties**] “The longest wavelength absorption band at $\lambda = 685$ nm (~~computed transition energies were scaled~~²⁸ by 72%²¹. ~~calibration factor of 0.72~~²⁸) was assignable to ...”

[Revised, References]

~~“Mainville, M., Ambrose, R., Fillion, D., Hill, I. G., Leclerc, M. & Johnson, P. A. Theoretical Insights into Optoelectronic Properties of Non-Fullerene Acceptors for the Design of Organic Photovoltaics. *ACS Appl. Energy Mater.* **4**, 11090–11100 (2021). Hashikawa, Y., Yasui, H., Kurotobi, K. & Murata, Y. Synthesis and properties of open-cage fullerene C₆₀ derivatives: impact of the extended π -conjugation. *Mater. Chem. Front.* **2**, 206–213 (2018).”~~

Fig. 3c Physical significance of the color coding should be explained (purple vs. blue vs. gray).

Our Response #17: In the caption, we have added the explanation of the color codes.

[Revised, Caption of **Figure 3c**] “c, LOL- π isosurface of **4'** (B3LYP-D3/6-31G(d,p); color codes, pink/blue for radially distributed π -orbitals and grey for π -orbitals which are not engaged in the caged π -surface).”

“the g_{abs} value of **4** is abnormally large on the ground of the effective radial π -conjugation along with the two [60]fullerene cages”

This is a key statement in the paper, but it must be supported with some justification.

Our Response #18: The corresponding paragraph was revised. The revised manuscript includes the discussion on comparison with monomer, singly-bonded dimer, and previously reported monomer fullerene.

[Revised, In the Section of **Chiroptical Properties**] “The similarly large values have been reported for hexahydropentalenone (ca. 0.2 supposedly at UV region)³⁵ and cycloarylene (0.167 at 443 nm)³⁶. Monomer **1** showed $g_{\text{abs}} = \pm 0.063$ at 648 nm, which is rather smaller than another open-[60]fullerene reported previously (± 0.20 at 710 nm)³². This might arise from the large 17-atom-ring orifice in **1** causing a flexibility of the caged structure, which partly breaks radial π -conjugation in solution while the latter is structurally rigid owing to the small 12-atom ring. As a result of an enhanced rigidity by dimerization, **5** attains a twice larger g_{abs} value (± 0.063 at 648 nm) than that of **1**. The full radial π -conjugation along with the two [60]fullerene cages further elevates the g_{abs} value in **4**. ~~Even when compared with monomer **1** ($g_{\text{abs}} = \pm 0.036$ at 652 nm) and non-conjugated dimer **5** ($g_{\text{abs}} = \pm 0.063$ at 648 nm), the g_{abs} value of **4** is abnormally large on the ground of the effective radial π conjugation along with the two [60]fullerene cages.~~”

“coalescence”
formation of one bond is hardly a “coalescence”

Our Response #19: We have revised the corresponding description.

[Revised, Conclusion] “The ~~chemical synthesis of the inter[60]fullerene conjugate coalescence of two open-[60]fullerenes by the organic synthesis~~ paves the way for the advent of fully-fused, double-opened C₁₂₀ nanocarbons with a well-defined structure.”

“fullerites”

Our Response #20: The term “fullerites” indicates polymeric coalesced fullerenes and it appears elsewhere. Ref. 8 also uses this term. So, we would like to keep this term for the introductory part with citing Ref. 8. On the other hand, we have revised our comment in the Conclusion.

[Revised, Conclusion] “This firmly stimulates the intellectual curiosity ~~of chemists and physicists~~ for applying them into chiroptoelectronic devices as well as for designing non-naturally occurring inter[60]fullerene allotropes that are otherwise inaccessible ~~conjugated open [60]fullerites as novel nanocarbon materials.~~”

“We anticipate that the science of fullerenes is now reborn to again stand in the spotlight so that we could synthesize and handle non-naturally occurring inter[60]fullerene allotropes that are otherwise inaccessible.”

This is a highly inappropriate final sentence.

Our Response #21: According to the comment from the Reviewer 4, we have removed the description in the revised manuscript.

[Removed, Conclusion] “~~We anticipate that the science of fullerenes is now reborn to again stand in the spotlight so that we could synthesize and handle non-naturally occurring inter[60]fullerene allotropes that are otherwise inaccessible.~~”